



**Atmospheric dry and wet nitrogen deposition in agro-pastoral**
**catchments of the China and Mongolia Altay**
Jin Ling Lv[1, 2, 3], Andreas Buerkert[4],Guo Jun Liu[1], Chao Yan Lv[1], Xi Ming Zhang[1], Kai Hui
Li[1]and Xue Jun Liu[1, 2*a]
[1]Xinjiang Institute of Ecology and Geography, Chinese Academy of Sciences,
Urumqi 830011, China;
[2]College of Resources and Environmental Sciences, China Agricultural University,
Beijing 100193, China;
[3]Institute of Plant Nutrition, Resources and Environmental Sciences, Henan Academy
of Agricultural Sciences, Zhengzhou, China;
[4]Universität Kassel, Organic Plant Production and Agroecosystems Research in the
Tropics and Subtropics, Steinstr. 19, D-37213 Witzenhausen, Germany
**Abstract**
Very few comparative studies of nitrogen (N) deposition in agroecosystems have been
conducted along landuse and altitude gradients. In an effort to fill this gap of
knowledge we selected three typical, interconnected landuse systems (cropland,
mountain grassland and plain grassland)at six sampling sites in the transboundary
Altay Mountains of NW China and SW Mongolia to compare the dynamics and
amounts of wet and dry N deposition. During 12 months from June 2014 to May 2015
dry and wet N deposition through middle volume total suspended particulates (TSP),
passive samplers and precipitation collectors were monitored. The croplands had the
highest concentrations of $NH_4^+$-N (1.6 mg N $L^{-1}$in China and 2.0 mg N $L^{-1}$in
Mongolia) and of $NO_3^-$-N (1.0 mg N $L^{-1}$in China and 1.2 mg N $L^{-1}$in Mongolia) in
precipitation compared with the other land use types for wet deposition. In contrast,
the Mongolian mountain grasslands experienced the highest wet deposition (3.2 kg N

[*a]Corresponding author (X. Liu), E-mailliu310@cau.edu.cn





ha$^{-1}$ yr$^{-1}$) which was at least partly due to higher summer precipitation (161 mm), the
second highest wet deposition occurred on Chinese cropland with 3.1 kg N ha$^{-1}$ yr$^{-1}$
while wet deposition in other landuse types ranged from 1.8 to 2.5 kg N ha$^{-1}$ yr$^{-1}$.
Chinese cropland had the highest NH$_3$ (3.1μg N m$^{-3}$) and NO$_2$ (3.8μg N m$^{-3}$)
concentrations and dry N deposition (15.3kg N ha$^{-1}$yr$^{-1}$) among all landuse types
while Mongolian cropland had dry N deposition of 8.9 kg N ha$^{-1}$yr$^{-1}$. Chinese
cropland (18.4 kg N ha$^{-1}$ yr$^{-1}$) had the highest total N deposition, followed by the
Mongolian cropland with 10.7 kg N ha$^{-1}$ yr$^{-1}$and the Mongolian mountain grassland
with 10.5 kg N ha$^{-1}$ yr$^{-1}$. NH$_4^+$-N concentration were negatively correlated with
precipitation (P<0.05). Concentration of NH$_3$ correlated positively with air
temperature (P<0.05) likely reflecting promoting effects of temperature on NH$_3$
emissions whereas NO$_2$ correlated negatively with temperature. Over all, croplands in
China had 72% higher N deposition than in Mongolia whereas the reverse was true
for mountain grasslands which received 31% more Nin Mongolia.
**Key words:** Agro-pastoral transition zone; Dry deposition; Land-use types;
Transborder watershed; Wet deposition





## Introduction


During the last three decades China´s industrial development and intensification of
agriculture and animal husbandry have greatly increased the concentration and
deposition of atmospheric reactive nitrogen (Nr; Liu et al., 2013).Since the 1980s
atmospheric Nr emissions have more than doubled in north and southeast China after
more than 30years of strong economic development (Liu et al. 2013) and current
atmospheric Nr deposition are very high (Pan et al., 2012; Xu et al. 2015). From the
North China Plain total nitrogen deposition of 54.4-103.2 kg N ha$^{-1}$ yr$^{-1}$have been
reported (Luo et al. 2013) and even for a remote oasis area in Xinjiang, northwest
China, N deposition of up to 35 kg N ha$^{-1}$yr$^{-1}$have been recorded (Liu et al. 2011; Li et
al. 2012).
There are several factors responsible for the increases in atmospheric N deposition
whereby agriculture and animal husbandry are two important sources (Aber et al.
1997; Granath et al. 2014; Simpson et al. 2014;Yang et al. 2010). The rapid
intensification of agriculture in response to the increasing demand for food has led to
enhanced use of mineral fertilizers in China, a key factor responsible for high regional
Nr deposition (Ju et al. 2009; Zhang et al. 2013). In addition, higher living standards
have increased the demand for meat and the development of intensive livestock
production systems (Huang et al. 2013; Wang et al. 2010).Overgrazing is also an
important problem that has been the subject of new landuse policies and even remote
agro-pastoral zones are affected by grazing pressure. This limits the productivity of
grasslands, depletes nutrients in grassland soils and may jeopardize the productivity of
animal husbandry systems.
Although some N deposition studies have been conducted in agricultural and
pastoral areas (Luo et al. 2003; Li et al. 2012; Du et al. 2014; Basto et al. 2015; Huang
et al. 2015; Liu et al. 2015; Wasiuta et al. 2015), little research has been conducted to
quantify atmospheric N deposition in the agro-pastoral zone. In China over 12
provinces comprising 160 towns and around 117 million people depend on this



landuse system which is mainly distributed in the northeastern and northwestern part
of the country and in Inner Mongolia (Du et al. 2009; Xu et al. 2014; Zhang et al.
2015). Monitoring of atmospheric dry and wet N deposition is therefore urgently
needed in this region.

The present study was conducted at the Altay Mountains, at the border area of

northwest China and western Mongolia representing a typical agro-pastoral transition
zone. To compare the effects of exposition and landuse intensity, we choose the same
agro-pastoral landuse type on the eastern Mongolian slope of the Altay Mountains.
The two nearby study areas have similar landuse types but different cropland/grazing
land ratios, intensities of mineral fertilizer input, types of animal husbandry systems
and levels of urban development. The combination of these factors may lead to
significant differences in N deposition across the year. Our aims were to quantify
seasonal variations in atmospheric N deposition and to compare the difference of dry
and wet N deposition in China and Mongolia agro-pastoral catchments.





## 2. Materials and methods

### 2.1. Sampling sites

The study was conducted in Qinghe county, northwest China and adjacent Bulgan county in Mongolia comprising a study area ranging from 45-47 °N and 89-91 °E. The topography is characterized by a gradual decline in elevation from north to south and is divided into high, intermediate and low mountains, hills, and the Gobi desert zone. The average altitude of Qinghe county is1218mabove sea level (a.s.l.), with a maximum elevation of 3659 m a.s.l. and a minimum of 900 ma.s.l.. Qinghe county is situated in the continental north temperate arid climatic zone, without four distinct seasons. Year round air humidity is very low with an average annual precipitation of 161 mm and an annual potential evaporation of 1495 mm. Winters are long and cold with an absolute minimum of -53 ℃ followed by cool and short summers with a recorded maximum temperature of 36.5 ℃ and an average air temperature of 0 ℃(Fig. 2). The pasture area in Qinghe comprises about 14300 km$^2$ and cultivated land amounts to 0.126 million hectares with 1.64 million livestock heads sent to market annually. Most of the agricultural area is planted to spring wheat (*Triticumaestivum* L.), but alfalfa (*Medicago sativa* L.) and sea buckthorn (*Hippophaerhamnoides* L.) are also widely grown. Wheat is sown in early May and harvested at the end of September. The amount of mineral N fertilizer applied to wheat ranges from 300-350 kg N ha$^{-1}$ yr$^{-1}$ in the Chinese croplands. Large numbers of sheep, cattle and camels are moved into the mountain grassland from July to September while during the winter months they remain for stubble grazing in the oasis croplands and desert margins.

Founded in 1931 Bulgan county (or Soum) neighbors Bayan-Olgii Province in the north, Northwest China in the east, and Uyench and Altay soum of Khovd province in the south. Its territory comprises 8105 km$^2$ at an average altitude of 1164 m a.s.l.. It has a continental climate with four seasons: April and May are the windiest months, January is the coldest (-40 ℃) and July is the warmest (+35 ℃) month of the year.



Annual precipitation averages 120-140 mm, with pronounced spring and autumn
seasons allowing some cultivation of arable crops.
The most of the cropland of Bulgan soum is also planted to spring wheat and to a
lesser degree to rye (*Secalecereale*L.), but less intensively with 150-250 N kg ha$^{-1}$ yr$^{-1}$.
Similarly to Qinghe, the growing season lasts from May to September and major
numbers of sheep, cattle and camels are moved into the mountain grassland from July
to September and spend the winters in the lowlands.
**2.2. Measurement of N deposition and analytical procedures**
From June 2014 to May 2015 wet (i.e. bulk) and dry N concentrations and deposition
were monitored and quantified at six sites in the border area of Altay Mountains (Fig.

1).

*2.2.1 Rainwater collection and calculation of wet N deposition*
Rainwater samples were collected with precipitation collectors directly after every
rainfall event in Chinese (CC) and Mongolian (MC) croplands by local farmers and/or
herdsman and the dates and amounts of rainwater were recorded. In addition, some
special rainfall collectors (three sub-samples per site) were employed at mountain
grasslands and plain grasslands in both China and Mongolia due to the difficulty in
collecting samples after every rainfall event. Each rain collector comprised a funnel
container 40 cm in diameter, a plastic hose and a 10-l plastic bucket. The funnel
container was set at a height of around 1.5 m above the ground to avoid dust or leaf
contamination from the ground surface. The plastic kettles were grounded at 30 cm
depth, stick out 5cm of the ground and were covered with a lid to prevent entry of
dust or other pollutants connected to the funnel with a plastic hose. Chloroform
(CHCl$_3$, 1-2 ml) was added to each bucket to inhibit N transformations in the
rainwater samples. The amount of precipitation was measured by an automatic
meteorological station nearby Meteorological Bureau of Qinghe Country. The
precipitation samples were collected manually once per month at the six sampling
points and transferred to plastic bottles followed by storage in a refrigerator at





-10 °Cuntil analysis. We also sampled snow at the beginning and the end of the
snowfall period and combined it with the snowfall data to calculate winter wet N
deposition. All samples were analyzed for $NH_4^+$-N and $NO_3^-$-N (inorganic N)
concentrations using an AA3 continuous flow analyzer (Seal Analytical Ltd.,
Southampton, UK). Wet deposition of inorganic N was calculated according to Luo et
al. (2014) as follows:
Wet N (every rainfall event, kg N ha$^{-1}$) = inorganic N concentration (mg N L$^{-1)}$
×precipitation (mm) ×0.01
Wet N (every month, kg N ha$^{-1}$) = 0.001 ×∑ N (every rainfall event or month)
*2.2.2 $NH_3$ and $NO_2$ collection and N calculation*
Atmospheric $NO_2$ was collected with passive samplers using Gradko diffusion tubes
from the UK Environmental Change Network (Goulding et al., 1998; Bush et al.,
2001). The $NO_2$ samplers consisted of polyethylenetubes (71.0 mm long and 11.0 mm
internal diameter), two caps, and stainless steel mesh disks. Two dry disks were
placed in the caps and 30 ml of a 20% aqueous solution of triethanolamine was
pipetted into the gray cap. The samplers were suspended at a height of 1.5 m (at least
0.5 m higher than the canopy height) above ground and exposed between15 days and
30 days in the air every month. The disks were extracted with a solution containing
sulphanilamide, $H_3PO_4$ and N-1-naphthylethylene-diamine dihydrochloride to
estimate the $NO_2$ concentration determined by colorimetry at a wavelength of 542
nm(Plaisance et al., 2004).
NH$_3$ samples were collected using ALPHA passive samplers (Adapted Low-cost
High Absorption, Center for Ecology and Hydrology, Edinburgh, UK). This
equipment included a tube, a plastic filter and a membrane (absorbed citric acid) and
was placedabout1.5 m above ground. The calculation was made according to Luo et al.
(2014) as follows:

V=DAt/L

Where t represents the time interval; D=2.09×10$^{-5}$m$^{-2}$s$^{-1}$ at 10°C, A=3.463×10$^{-4}$m$^{-2}$,



L=0.006m. The following equation was then derived:

$V(m^3)= 0.004343363(m^3) \times t(h)$

The concentration of $NH_3$ ($\mu g\ N\ m^{-3}$) was obtained as follow:

$C= (m_e-m_b)/V$

Where $m_e$ represents the amount of $NH_3$ in the experimental sample and $m_b$ represents
the amount of $NH_3$ in the blank sample.
*2.2.3  $pNH_4^+$ and $pNO_3^-$*
Airborne $PM_{10}$ particles (particulate matter whose aerodynamic equivalent diameter is
< 10 μm) were sampled using a middle flow particulate sampler (Tian hong
Instruments Co. Ltd., Wuhan, China) with a flow fluxes of 1.05 $m^3\ min^{-1}$, and 7-10
daily samples of $PM_{10}$ were collected at QC and BC during each month. Samples at
other sites were not taken given lacking power and harsh environmental conditions.
The membrane of $PM_{10}$ was glass fiber and it was placed in an incubator at constant
temperature and humidity (22°C, relative humidity 50%) for 24 hours before and after
sampling and weighed on an electronic balance. Finally, the samples were placed in
beakers containing 50ml ultrapure water and ultrasonicated for 30 min. The extracts
were filtered through 47-mm Whatman GF/F membrane syringe filters (GE
Healthcare Bio-Sciences, Pittsburg, PA, USA). The filtrates were stored refrigerated
at 4 °C. Ammonium and nitrate in $PM_{10}$ ($pNH_4^+$ and $pNO_3^-$) were measured using a
Seal AA3 continuous flow analyzer (Seal Analytical Ltd., Southampton, UK).
**2.3 Estimation of dry N deposition**
Data on dry N deposition are complicated to collect given the effects of variable
weather conditions and differences in vegetation types. We did not use
micro-meteorological methods because of unavailability of equipment to measure dry
N deposition. Rather we estimated dry N deposition by multiplying the measured
concentrations of Nr species by their deposition velocities ($V_d$) obtained from related
studies published in the literature (Shen et al. 2013; Yu et al. 2014). The following
equations were used:



$F = C \times V_d$
Where by $V_d$ can be expressed by
$V_d = (R_a + R_b + R_c)^{-1}$
Where $R_a$ is the aerodynamic resistance, $R_b$ is the quasi-laminar boundary layer
resistance, and $R_c$ is the surface or canopy resistance (Shen et al. 2009). Because we
did not measure $V_d$, the $V_d$ values of the Nr species under different landuse types were
obtained from Flechard et al. (2011) for simplification.
**2.4 Statistical analysis**
Linear regression was used to analyze interactions among the different Nr species. For
Pearson $\acute{s}$ correlation and linear regression analyses, significance was defined at
$P < 0.05$. T-tests were employed to compare N deposition among monitoring sites,
land-use types and seasons. All statistical analyses were performed using the SPSS
18.0 software package (SPSS Inc., Chicago, IL, USA). Figures were prepared using
the Origin 8.0 software package (Origin Lab Corporation, Northampton, MA, USA).
**3 Results**
**3.1 Wet deposition of $NH_4^+$-N and $NO_3^-$-N**
The Mongolian cropland (MC) had the highest $NH_4^+$-N concentration in the wet
deposition compared with the Mongolian Mountain grassland (MM) and the
Mongolian Plain grassland (MP)(Table 2, Fig. 3).The $NH_4^+$-N concentrations of the
samples from Chinese sites were relatively low compared with the Mongolian
sampling sites. Chinese Cropland (CC) had are relatively high $NH_4^+$-N concentration
compared with the other two sampling sites in the Chinese Mountain grassland (CM)
and the Chinese Plain grassland (CP). $NO_3^-$-N concentrations were highest for CP in
China, followed by MC in Mongolia.

The different landuse types had different $NH_4^+$-N and $NO_3^-$-N peaks. Highest

cropland $NH_4^+$-N occurred in May in China and in September in Mongolia and the
$NO_3^-$-N peak occurred from March to May in China and from August to October in
Mongolia (Fig.3). The two countries had similar mountain pasture (CM and MM)



$NH_4^+$-N concentration peaks. Both occurred from July to September, and $NO_3^-$-N
peaks were recorded from July to September at QP and BM. The Chinese and
Mongolian Plain grasslands (CP and MP) had different $NH_4^+$-N concentration
dynamics, CP had a low value with no clear peak throughout the year while MP had a
significantly higher $NH_4^+$-N concentration and a peak occurring from June to
September. However, $NO_3^-$N showed the opposite trend, with CP having a
significantly higher peak concentration from June to October and MP having its
maximum value in June and similar values in other months (Fig. 4).
**3.2 Net $NH_3$ deposition concentrations**
Cropland had the highest $NH_3$ concentrations of all land use types whereby CC had a
maximum $NH_3$ value of 7.41μg N $m^{-3}$ in May and an average of 3.1μg N $m^{-3}$
throughout the rest of the year. The mountain grasslands (CM and MM) had the
lowest $NH_3$, with average concentrations of 1.07and 1.08μg N $m^{-3}$, respectively.
Plains grasslands had $NH_3$ concentrations during the key growing season (June to
October) of 1.53μg N $m^{-3}$ at CP and 1.94μg N $m^{-3}$ at MP(Table 2).The $NH_3$ values
during the growing and non-growing seasons were significantly different (P=0.008).
With exception of MM, significantly higher $NH_3$occurred during the growing season,
especially at the croplands (P=0.026) and the mountain grassland had a significantly
higher air $NH_3$ during the non-growing season (Fig. 5).
**3.3 $NO_2$ concentrations**
QC had the highest $NO_2$ concentration with an average value of 3.8μg N $m^{-3}$over the
year and a maximum value of 8.1μg N $m^{-3}$in June. MC had lower $NO_2$ with an
average value of 2.4μg N $m^{-3}$over the year. MM had a significantly higher $NO_2$ (2.6μg
N $m^{-3}$) than CM (1.6μg N $m^{-3}$). The CP grassland had a slightly higher $NO_2$
concentration of 2.2μg N $m^{-3}$ in the key growing season (June to October) than did
MP with 1.5μg N $m^{-3}$ (Table 2).The $NO_2$ values in the growing and non-growing
seasons were significantly different (P<0.001) for CC and MM. However, $NO_2$
concentrations were similar for the CM grassland and MP (P>0.322; Fig. 6).



**3.4 Particulate Nr species in the air**

Because of power and equipment limitations we chose the croplands in both countries (CC and MC) as the monitoring points for particulate Nr. The monthly $pNH_4^+$ concentrations were 0.75and 0.53μg N m$^{-3}$ for CC and MC, respectively (Table 2). The CC had a significantly higher $pNH_4^+$ concentration than the MC (P=0.033).The $pNH_4^+$ concentration peaked from July to August and the highest value (2.66 μg N m$^{-3}$) was attained in July (Fig.7). Monthly $pNO_3^-$ concentrations were 0.37μg N m$^{-3}$ at CC and 0.11μg N m$^{-3}$ at MC (Table 2). The CC had a significantly higher $pNO_3^-$ concentration than the MC (P=0.008) with peaks from July to August and April to May and a maximum value of1.38μg N m$^{-3}$in May.

In addition, the growing season had higher $pNH_4^+$ concentrations, especially in CC. Average $pNH_4^+$ concentrations of CC were 60% higher than those of MC. For $pNO_3^-$ concentrations values were similar between growing season and non-growing season in both countries (P=0.302). However, average $pNO_3^-$ concentrations of CC was three times higher than for MC (Fig.8).

**3.5 Wet, dry and total N deposition**

Annual wet N deposition amounted to 2.0-3.1 kg N ha$^{-1}$ yr$^{-1}$at the Chinese sites and 1.8-3.2kg N ha$^{-1}$ yr$^{-1}$ at Mongolian sites. Among the six sampling sites, the highest wet deposition occurred at the MM and CC reflecting high precipitation or high $NH_4^+$-N and $NO_3^-$-N concentration, with values of 3.1 and 3.2 kg N ha$^{-1}$ yr$^{-1}$ for the Mongolian and Chinese sites, respectively. Wet Deposition was smallest at MC given lowest precipitation. Wet deposition rates at other sites fell in-between. The CC had the highest N dry deposition rate (15.3 kg N ha$^{-1}$). The second was the MC with 8.9 kg N ha$^{-1}$. The MM grassland had a higher dry deposition (7.3kg N ha$^{-1}$) than its Chinese counterpart (5.5 kg N ha$^{-1}$). Dry deposition rates in plain grasslands were similar across countries.

Total N deposition in CC was 72%higher than in MC, but MM grassland had a higher total N deposition than CM grassland. The MP grassland had a similar total N



deposition (7.4 kg N ha$^{-1}$) than CP grassland (7.7 kg N ha$^{-1}$). The wet N deposition
species ($NH_4^+$ and $NO_3^-$) altogether accounted for 16.9-31.2% at the Chinese sites and
for 16.8-30%at the Mongolian sites. Dry N deposition accounted for 69-83% at the
Chinese sites and 70-83% at Mongolian sites.
**4 Discussion**
**4.1 Methodological evaluation of dry and wet deposition collection**
Due to the difficult infrastructural conditions (long distances, high altitude and
cross-border problems), compromises needed to be made with respect to sampling
equipment and collection intervals. While the self-made equipment for rainfall
sampling was established at six sites (Fig 3) from June 2014 to May 2015, for the
cropland and mountain grassland, we collected rainfall or snow samples every month
across the year. For the plain grassland, however, we just collected samples every
month during the growing season. Outside the growing season samples were collected
just for the first time and last snowfall event to compute average values. The middle
flow particulate sampler was just established in the cropland given human
surveillance there but not in the mountain and plain grassland where power was
lacking For the $NO_2$ and $NH_3$, we collected the samples 20 days one time in six
sampling points. The cropland and mountain grassland samples were collected from
June 1, 2014 to May 31, 2015, and the plain grassland samples were just collected in
growing season due to harsh environmental conditions.

In addition the use of the passive sampling devices may have led to severe

underestimations of deposition given the hyper arid conditions of our study zone.
Relative air humidity was at around 30% in summer and 80-90% in winter, and
lowest winter temperatures were -40℃ in winter. Under these conditions the $NH_3$
(ALPHA) and $NO_2$ samplers were to the best of our knowledge never tested before.
Similar studies show that Palmes $NO_2$ diffusion tubes (Gradko 7.1 cm open diffusion
tubes) may be used over a temperature range from -50℃ to 40℃and a relative
humidity range from 30% to 95% (Bush et al. 2001; Plaisance et al. 2004; Gerboles et





al., 2005). For the ALPHA samplers the hygroscopic nature of the citric acid may
allow for reliable measurements even at 30-35% RH (Perrinoet al., 2002). However,
our data certainly merit methodological verification under laboratory and field
conditions.
**4.2 Atmospheric dry and wet N deposition**
Dry deposition includes gas emissions and particulate Nr deposition (Shen et al. 2013;
Granath et al. 2014; Maaroufi et al. 2015). In our experiment CC had significant
higher $NH_3$ and $NO_2$ concentration than the other landuse types, mainly due to the
large area of cropland on the Chinese side of the border, together with the excessive
inputs of mineral fertilizer N, which likely led to large losses *via* $NH_3$ volatilization
and soil NOx emissions.

The Chinese cropland also had the highest inorganic N concentrations and thus

had higher dry deposition than the Mongolian cropland. Moreover, cropland had
higher dry deposition than the other landuse types. There were usually higher $NH_3$
emissions in the growing season, mainly due to the fertilizer or manure applications
during the growing season. The MM grassland had higher $NO_2$ depositions than the
CM grassland, presumably because the Mongolia site had many herdsmen living in
the area over most of the year and, especially in winter, large amounts of coal, wood
and cattle manure are burned for home heating from October to May. Many of the
herdsmen in China move to the mountains only from July to mid-September in
summer, with very few people living there during the winter. Similar conclusions hold
for the wet deposition.

The monthly concentrations of $NH_3$ showed significant positive correlations with

temperature (P=0.009) but no correlation with RH (P=0.491) or $NO_2$ (P=0.580;
Fig.10). A similar trend was also found in Guangzhou in south China and in an
agricultural catchment in subtropical central China (Ju et al. 2009; Shen et al. 2013).
This indicates that increasing temperature promotes the emission of $NH_3$. Gaseous
$NO_2$ was also positive correlated with temperature (P=0.018) but not with RH or $NH_3$





339 (P=0.153).This conclusion is consistent that of with Luo et al. (2013) who studied dry

340 deposition in northern China. This may also imply that $NO_2$ emissions mostly occur

341 as a consequence of human activities, especially the combustion of fossil fuels and

342 automobile exhausts.

343  The amount of rainfall had a significant effect on the concentration of inorganic N.

344 The higher amount of precipitation, the lower the inorganic N concentration (Fig.9),

345 especially in the case of $NH_4^+$ which was significantly correlated with the

346 precipitation (P=0.039). $NH_4^+$ and $NO_3^-$ werenot significantly correlated with one

347 another (P=0.143), which indicated that the results for the wet deposition are greatly

348 influenced by the dry deposition. All in all, the different landuse types did not differ

349 significantly in their wet deposition in either country.

350 **4.3 The uncertainty of the compensation point between the $NH_3$ emission and**

351 **deposition in three landuse styles**

352 The concentration of $NH_3$ in the air is susceptible to be affected by meteorological

353 and anthropogenic factors. On the one hand, part of atmospheric ammonia settled onto

354 the soil surface, on the other hand, part of $NH_3$ volatilize from the surface soil.

355 Therefore, it is very hard to accurately estimate net $NH_3$ deposition under our

356 conditions. In order to better estimate the $NH_3$ deposition value, it is common practice

357 to calculate the deposition velocity rate by means of meteorological factors to get the

358 appropriate deposition compensation point. In our experiment, the landuse styles

359 included alpine meadow, plain grassland and farmland. In the farmland, 5.0µg N m$^{-3}$

360 was assumed as the compensation point of dry deposition of $NH_3$ in the growing

361 season (Shen et al. 2013), and 0 µg N m$^{-3}$ was assumed as the compensation point of

362 dry deposition of $NH_3$ in the no-growing season due to low $NH_3$ volatilization. In the

363 mountain and plain grassland, 0 µg N m$^{-3}$ was chosen as the compensation point of

364 dry deposition of $NH_3$ due to low $NH_3$ volatilization (Li et al., 2012; Shen et al. 2013).

365 In addition we observed in our study area that N deposition was spatially very

366 unevenly distributed, particularly between mountain pastures and plain pastures.



Nitrogen deposition was possible higher next to herdsmen's houses, roads or
sheepfolds due to more pronounced $NH_3$ or $NO_X$ releases. Farm- and grasslands are
intertwined in our research areas. Therefore, much uncertainly for wet and dry N
deposition remain.

**5 Conclusions**
The agro-pastoral area around Qinghe (China) and Bulgan (Mongolia) differed in
atmospheric N deposition across landuse types. The mountain grasslands had
relatively higher wet deposition reflecting much higher rainfall and Nr emissions.
Chinese croplands had higher wet and total N deposition than Mongolian croplands
due to higher population and chemical fertilizer input, but higher N deposition were
found in the Mongolian mountain grassland than Chinese mountain grassland due to
different grazing systems. Nearly all land use types had higher N deposition in the
(warm) growing season than the in the winter months. Compared with Mongolia,
Chinese grassland faces more pronounced Nr losses due to additional N deposition
and overgrazing. Thus, it is necessary to reduce the application of N-fertilizers to
croplands as well as herd numbers.

**Acknowledgements**
We acknowledge Dr. Peter Christie (UK) for his valuable comments and linguistic
corrections of the manuscript. We also thank Dr.Sven Goenster (Universität Kassel,
Germany) for his contribution of meteorological data and sample collection. The
study was supported by the WATERCOPE (I-R-1284-WATERCOPE) project funded
by IFAD (International Funding for Agriculture Development, Rome, Italy), the State
Basic Research Program (2014CB954200) and the Chinese National Natural Science
Foundation (41425007, 31421092).






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



**Table 1** Description of the six sampling sites in the Chinese and Mongolian Altay
Mountains.

| | Site | Landuse | Latitude | Longitude | Elevation (masl) | Annual mean temperature (℃) | Annual precipitation (mm) | Sampling period |
|---|---|---|---|---|---|---|---|---|
| China | Qinghe (CC) | Cropland | 46 °44' | 90 °19' | 1126 | 5.4 | 123 | Jun 2014 -May 2015 |
| | Huzert (CM) | Mountain grassland | 46 °40' | 90 °24' | 1605 | 1.2 | 149 | Jun 2014 -May 2015 |
| | Guojiazhan (CP) | Plain grassland | 46 °08' | 89 °58' | 1284 | 4.3 | 78 | Jun 2014-Oct 2015 |
| Mongolia | Bulgan Sum (MC) | Cropland | 46 °6' | 91 °34' | 1184 | 3.9 | 56 | Jun 2014 -May 2015 |
| | Turgen (MM) | Mountain grassland | 46 °49' | 91 °21' | 1889 | -0.5 | 161 | Jun 2014 -May 2015 |
| | Bayangol (MP) | Plain grassland | 46 °20' | 91 °25' | 1323 | 4.2 | 83 | Jun 2014 -Oct 2015 |













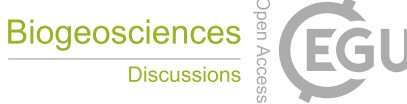


**Table 2** Annual volume-weighted mean concentrations of $NH_4^+$-N and $NO_3^-$-N in

rainwater and the annual mean concentrations (standard deviation) of gaseous and

particulate Nr species in the air at the six sampling sites in the Chinese and Mongolian

Altay Mountains.

| Site | $NH_4^+$-N mg N L$^{-1}$ | $NO_3^-$-N mg N L$^{-1}$ | $NH_3$ µg N m$^{-3}$ | $NO_2$ µg N m$^{-3}$ | $pNH_4^+$ µg N m$^{-3}$ | $pNO_3^-$ µg N m$^{-3}$ |
|---|---|---|---|---|---|---|
| CC | 1.6 (0.2-6.2)* | 1.0 (0.4-2.1) | 3.1(1.8-7.4)$^b$ | 3.8(0.9-8.1) | 0.8(0.3-2.8)$^a$ | 0.4(0.1-1.38)$^a$ |
| CM | 1.0 (0.1-3.1) | 0.7 (0.1-1.2) | 1.1(0.3-2.3) | 1.6 (0.1-2.6) | | |
| CP | 0.6 (0.3-0.8) | 2.0 (1.4-3.2) | 1.5(0.3-2.8) | 2.2(1.8-3.3) | | |
| MC | 2.0 (0.2-5.9) | 1.2 (0.5-2.0) | 1.7 (0.9-3.3) | 2.4(0.6-5.8) | 0.5(0.1-1.2)$^b$ | 0.1(0.03-0.43)$^b$ |
| MM | 1.2 (0.4-5.5) | 0.8 (0.3-1.7) | 1.1(0.6-1.8) | 2.6(0.2-5.5) | | |
| MP | 1.8 (0.3-3.7) | 0.7 (0.2-3.4) | 1.9(1.1-2.8) | 1.5(0.2-3.1) | | |

*Values in the parentheses indicate the variation range of the Nr of the rain across the whole year.

[a, b] Different letters within the same column indicate statistical differences in variables mean among landuse types as shown by

the Tukey's multiple range test (P<0.05).



**Table 3** Wet and dry N deposition (kg N ha$^{-1}$ yr$^{-1}$) at the sampling sites in the Chinese
and Mongolian Altay Mountains from June 2014 to May 2015.

|  | Site | Rainfall (mm) | Wet deposition | | Dry deposition[a] | | | | WD[b] | DD | TD |
|---|---|---|---|---|---|---|---|---|---|---|---|
|  |  |  | NH$_4^+$ | NO$_3^-$ | NH$_3$ | NO$_2$ | pNH$_4^+$ | pNO$_3^-$ |  |  |  |
| China | QC | 123 | 1.91 | 1.23 | 7.3 | 7.1 | 0.6 | 0.3 | 3.1 | 15.3 | 18.4 |
|  | QM | 149 | 1.49 | 1.04 | 2.5 | 3.0 |  |  | 2.5 | 5.5 | 8.0 |
|  | QP | 78 | 0.47 | 1.56 | 3.6 | 4.1 |  |  | 2.0 | 7.7 | 9.7 |
| Mongolia | BC | 56 | 1.12 | 0.67 | 3.9 | 4.5 | 0.4 | 0.1 | 1.8 | 8.9 | 10.7 |
|  | BM | 161 | 1.93 | 1.29 | 2.5 | 4.8 |  |  | 3.2 | 7.3 | 10.5 |
|  | BP | 83 | 1.49 | 0.58 | 4.6 | 2.8 |  |  | 2.1 | 7.4 | 9.5 |

[a] Dry deposition velocities of NH$_3$, NO$_2$ were 0.74 and 0.59, respectively, as cited from Shen et al.(2011)
[b] WD: total wet N deposition, DD: total dry N deposition, TD: total N deposition






















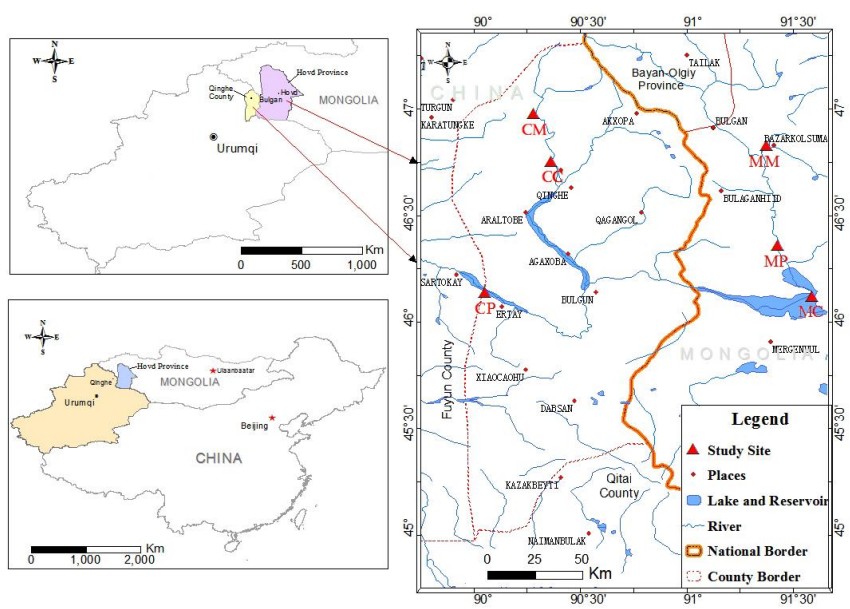


**Fig.1.** Map of the six sampling sites in the agro-pastoral catchment of the Chinese and
Mongolian Altay Mountains.










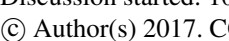



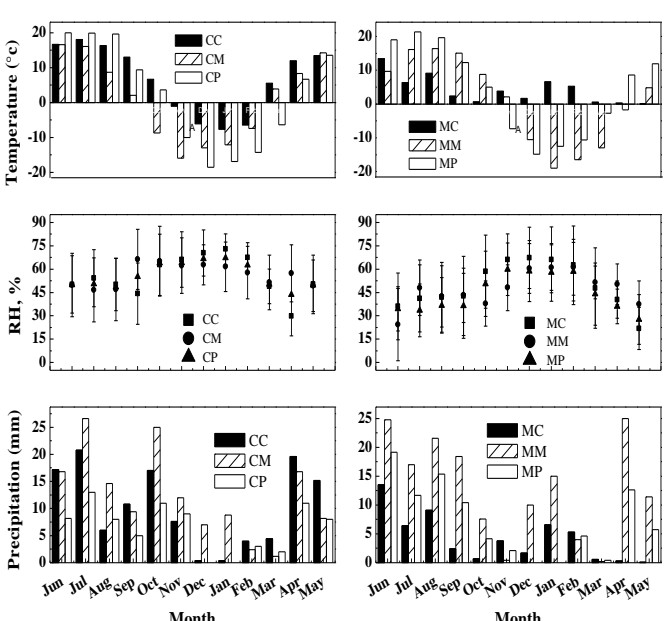


**Fig. 2.**Monthly mean air temperature and relatively humidity (RH) at six sampling

sites of the Chinese and Mongolian Altay Mountains.


















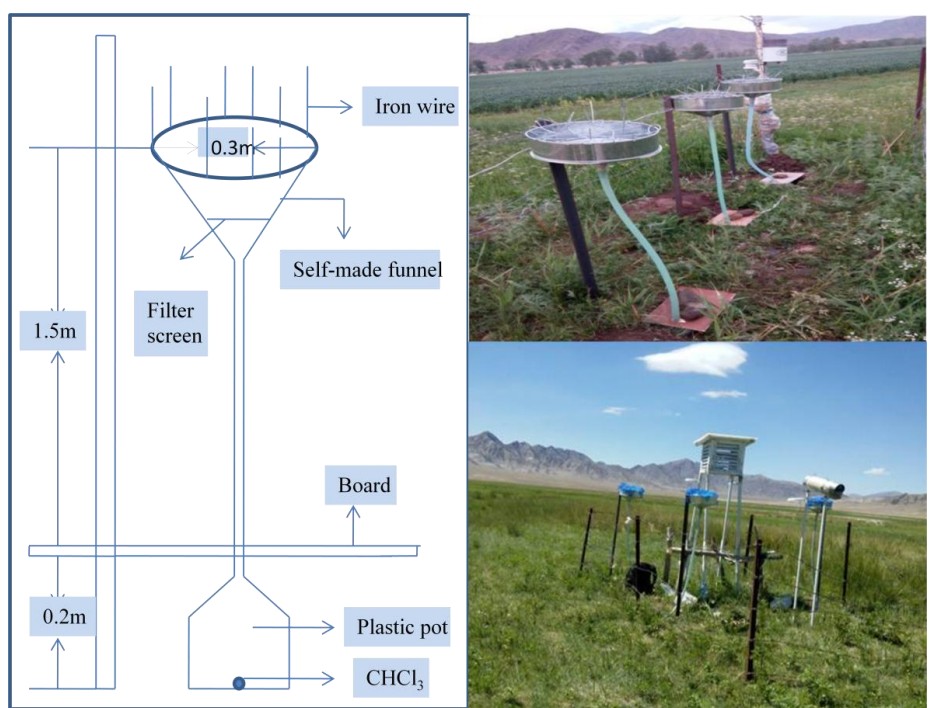


**Fig. 3.** The self-made wet collection equipment at the sampling sites in the Chinese

(up right) and Mongolian Altay Mountains (down right)














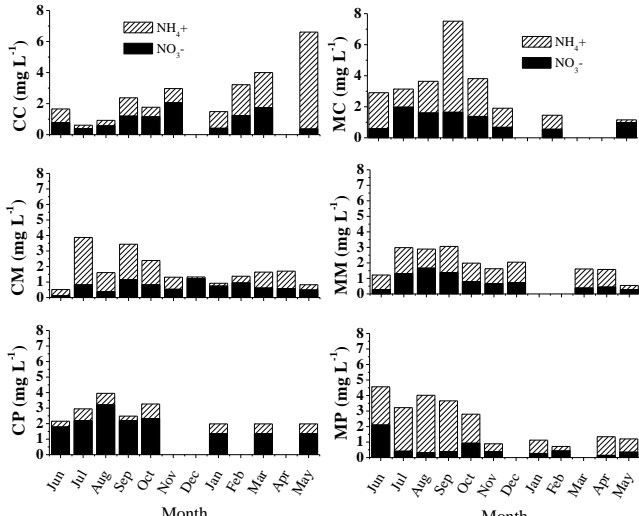


**Fig.4.**Concentration of $NH_4^+$-N and $NO_3^-$-N of wet deposition at six samples sites in
the Chinese and Mongolian Altay Mountains

















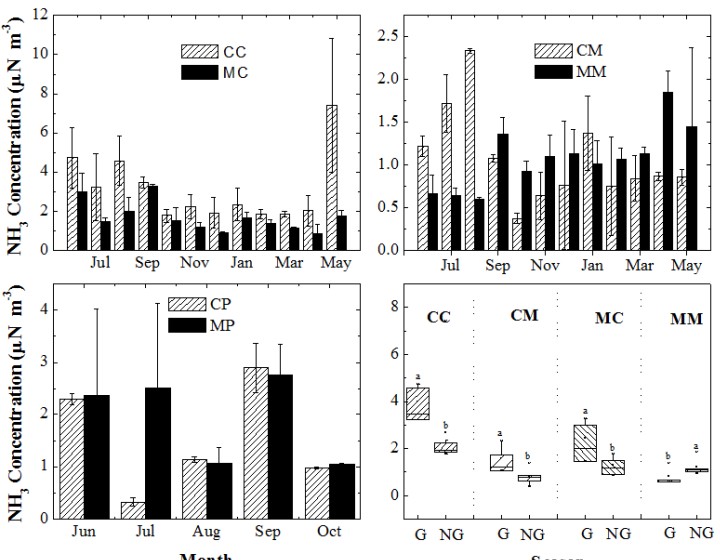


**Fig.5.**Monthly concentrations of NH$_3$-Nin the growing season (G) and the
non-growing season (NG) at six sites in the Chinese and Mongolian Altay Mountains












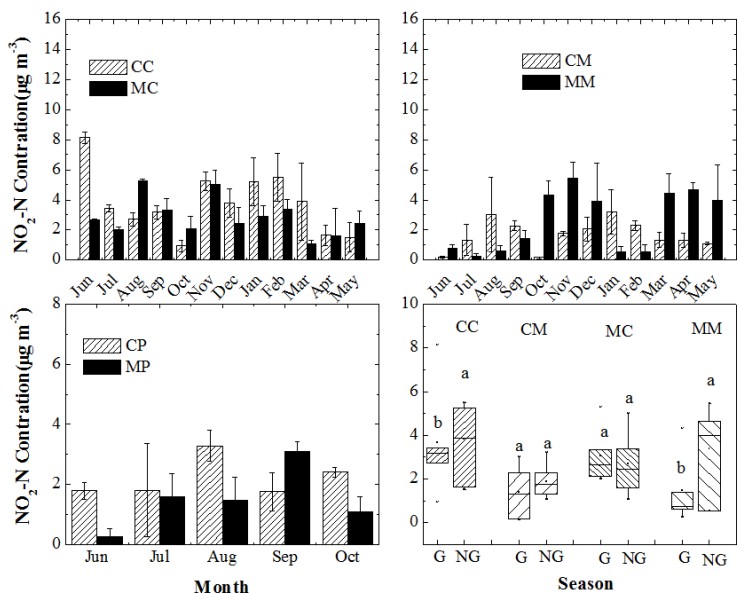

**Fig.6.**
Monthly concentrations of NO$_2$-N in the growing season (G) and the non-growing
season (NG)of six sites in the Chinese and Mongolian Altay Mountains












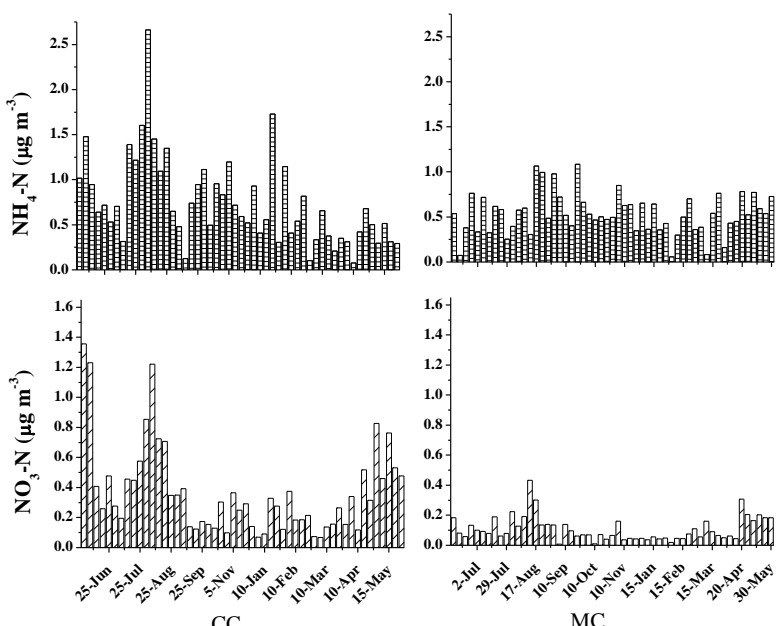



**Fig.7.** Monthly concentrations of NO$_2$-N at six sampling sites in Chinese and
Mongolian Altay Mountains





















**Fig.8.**Concentrations of $NO_2$-N in the G (growing season) and the NG (non-growing
season) at six sites in the Chinese and Mongolian Altay Mountains





















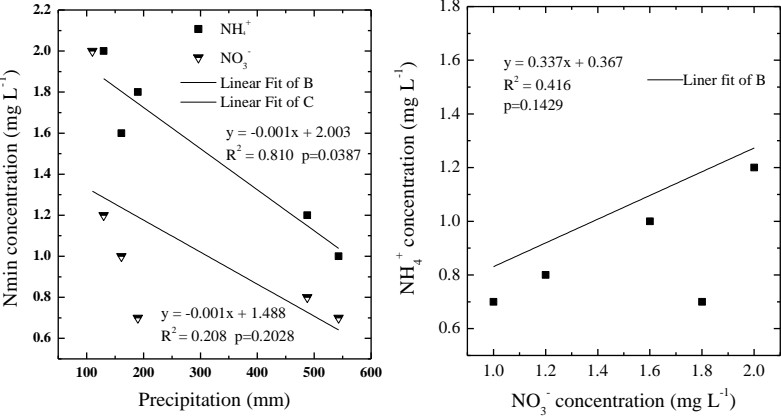


**Fig.9.** Relationship between monthly precipitation and $NH_4^+$-N and $NO_3^-$-N in

rainwater at six sampling sites in the Chinese and Mongolian Altay Mountains


















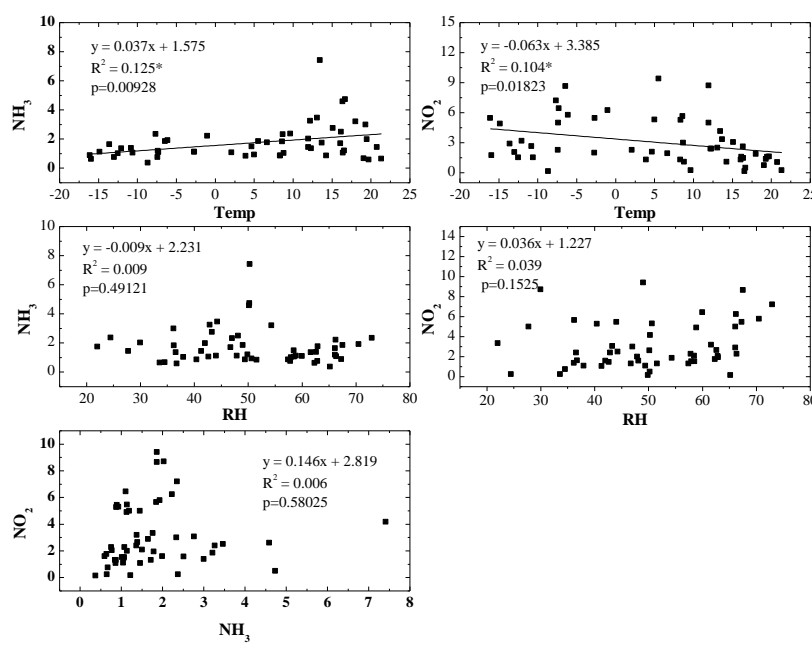


**Fig.10.** Relationship between atmospheric $NH_3$ and $NO_2$ and temperature (Temp) and
relatively humidity (RH) in the Chinese and Mongolian Altay Mountains.




