# Peer review of "Atmospheric dry and wet nitrogen deposition in agro-pastoral"

_Biogeosciences, 2017_

## Referee Comment (RC1) · Anonymous Referee #2 · 5 May 2017

The study reports atmospheric dry and wet deposition in agro-pastoral catchments in the Chinese and Mongolian Altay, where such data are limited. Although the study is interesting and publishable, many issues exist in the current version of the manuscript. I suggest the authors to carefully check the manuscript and substantially improve it. Major issues include: (1) The introduction section is not very related to the topic of the manuscript. As mentioned by the authors, lots of studies have been conducted in both agricultural and pastoral areas. Why are studies in agro-pastoral zones needed considering lots of studies have been conducted in both agricultural and pastoral areas? What are the hypotheses of the current study? (2) There are still lots of language and punctuation issues which reduces the readability of the manuscript. Which should be

resolved before it can be published. In addition, the style of the presentation leads to confusion. For example, "The Mongolian cropland (MC) had the highest NH4+-N concentration in the wet deposition". In my understanding, it should be "The NH4+-N concentration in rainwater collected over Mongolian cropland was highest". The authors should check such issues throughout the manuscript. (3) The site description and management of the agricultural/pastoral fields are not clear. Where are croplands or pasture distributed over the region, e.g., at what latitude range? What does "mountain grassland and plain grassland" mean? For both croplands and pastures, how many fertilizers are applied? How about the intensity of pasture management. And such on. (4) The discussion needs substantial improvement. Most of the section is only repeating the results or should be put into the result section. I suggest the authors focus on (1) comparison with other studies, (2) exploring the mechanisms underlying the difference in N deposition among the land use types, and (3) discussing the implications of the observations.

Other issues include: (1) Throughout the manuscript, "landuse" should be changed to "land use". (2) What does "mountain grassland and plain grassland" mean? (3) Line 19: there should be a space between ")" and "at". (4) Line 20: Should explain NW and SW in their first appearance. I don't think there is need to use abbreviation in these cases. Furthermore, (5) Line 26: delete "for wet deposition" since these data are for precipitation as mentioned before. (6) Lines 27-30: please make sure whether there is significant difference among these rates. If the difference is not significant, you can't say which is the highest, or higher than another. (7) Lines 31-33: It is hard to imagine that the location with the highest concentrations were not accompanied by the highest dry deposition. (8) Line 36: NH4+-N concentration in the precipitation? Please specify. It should be was not were. (9) Line 39: Soil should be the major source of atmospheric NO2 in such remote region. Normally soil emission of NO2 increases with temperature until an optimum temperature (close to 30 °C?) is reached. So how to explain the negative relationship between temperature and atmospheric NO2 here? Line 39: not "Over all", but "Overall". Line 41: . . .N in. . .. Line 70: "little research has"

should be "few researches have". Line 128: Only at croplands? Lines 127-142: There is confusing. You mentioned that rainwater samples were collected after every rainfall (lines 127-128). However, "the precipitation samples were collected manually once per month" (Line 141). Which is the truth. If the collectors were exposed in the field for one month, both wet and part dry deposition should be included, right? Line 177: Should be "Particulate ammonium and nitrate" Lines 209-210: Why not use one way analysis of variance? Line 215 and others: should be NH4+-N concentration in the rainwater, not in the wet deposition. For deposition, the unit should be a rate, e.g., kg N ha-1 yr-1. Line 235: Similar to the above comment. I can't understand what it means by saying "Net NH3 deposition concentrations". I suggest to use "Atmospheric NH3 concentrations". Line 246: Atmospheric NO2 concentrations? Lines 292: Better change "established" to "deployed". Lines 291-294: please check the sentence. Lines 289-314: For most of this section, you are only repeating the information which should be moved into the method section. Line 316: Dry deposition includes gas emissions? Lines 350-351: The compensation points between the NH3 emission and deposition? Line 2: Chinese and Mongolian Altay Fig 8. The subscripts and superscripts are not normal. There are too many figures. I suggest to put some as the supplemental materials.

---

## Referee Comment (RC2) · Anonymous Referee #1 · 8 May 2017

The manuscript (BG-2017-55) reported filed measurement of atmospheric nitrogen dry and wet deposition in different land uses in China and Mongolia. The chosen topic is interesting but the manuscript is not well written and there are some concerns about the methodology and discussions.

Some scientific issues. (1) The experiment was conducted only for one year, which is not enough for a filed observation of nitrogen deposition. Nitrogen deposition is influenced by emission, weather conditions, which have high variation among years. The manuscript try to estimate annual N deposition in several land uses by one year observation, the results are not representative. (2) The study measured gases and particulate N concentration and use dry deposition velocity Vd to calculate nitrogen

dry deposition. But authors choose Vd values from a reference (Flechard et al, 2011). And from Table 3, authors use same Vd of NH3 and NO2 for all three canopy and two countries. This methodology is not valid. Vd is affected by weather conditions and canopy types. It is changed diurnal, seasonal, and yearly for different plant types in different places. Therefore, it is not proper to European Vd data to represent Vd in China and Mongolia. Since weather data are available for authors from meteorological stations, Vd (hourly, daily or monthly) for different canopy of different Nr could be calculated easily. There are many widely used methods for this calculation, such as using big-leaf model. (3) This study used the self-made wet collection equipment as shown in Fig. 3. However, the equipment is not well designed. The top of the equipment is open without a lid, which may also collect some dry deposition especially particulate N deposition, therefore, the collected deposition by this kind of equipment is usually considered as a mixture of wet deposition and part of particulate dry deposition. By using this equipment, the wet deposition is therefore overestimated. Authors did not include any correction for this either, which resulted in high uncertainty of the reported N deposition amount. (4) The discussion needs improve. All discussion section just repeat some results and simply compared with several other studies but no detailed and deep discussion was presented. The difference of total N deposition amount and contribution of different Nr forms for different land uses could be due to the difference of fertilizer input, management, emission, and deposition process. And the difference between two countries might reflect the difference of management tradition and urban development. The discussion about Nr deposition and their potential sources among different land uses and between China and Mongolia are much more important and interesting than the comparison of the annual values with other studies since one year observation result in the manuscript is not enough to represent annual N deposition. (5) The section 4.3 try to discuss the uncertainty of N deposition, but authors just discussed about NH3 compensation point. It will be better if authors could add an uncertainty analysis about N deposition, which could partly compensate the shortage of one year observation.

There are many format and punctuation problems in the manuscript. I suggested authors to do some technical corrections about spelling and format at the Initial Manuscript Evaluation report, but unfortunately, these errors were not corrected for the discussion paper. (1) Punctuation problems. Space is needed after each word! For instance, page 1, line 19, add a space before "at six sampling…"; page 1, line 24, add spaces before "in China" and before " in Mongolia". There are lots of this problem in the manuscript. Please carefully check the whole manuscript and corrected these problems! (2) Table 3, reference Shen et al. (2011) was not found in References. Please check citation and reference list. (3) Fig.1, please add Nansha Islands for China map. (4) Fig. 6, figure caption should start from the next line. (5) Fig. 7 and Fig. 8, figure captions are not consistent with figures. (6) Fig. 9, remove "liner fit of B" in figure.

---

## Author Comment (AC2) · 31 May 2017

The manuscript (BG-2017-55) reported filed measurement of atmospheric nitrogen dry and wet deposition in different land uses in China and Mongolia. The chosen topic is interesting but the manuscript is not well written and there are some concerns about the methodology and discussions. Some scientific issues. (1) The experiment was conducted only for one year, which is not enough for a filed observation of nitrogen deposition. Nitrogen deposition is influenced by emission, weather conditions, which have high variation among years. The manuscript try to estimate annual N deposition in several land uses by one year observation, the results are not representative. Response: Thanks for your suggestion, the more year's experiment, the better result for

the N deposition. However, due to the difficulty to collect the sample, for example, we often treked hundreds of miles from mountain to plain, and we also need to cross the boundary to Mongolia to carry out relevant tests. Although we know that there may be some differences for atmospheric dry and wet deposition in different year, we were as much as possible to arrange well the experiment to ensure every points can represent the characteristics of the region for atmospheric dry and wet deposition.

(2) The study measured gases and particulate N concentration and use dry deposition velocity Vd to calculate nitrogen dry deposition. But authors choose Vd values from a reference (Flechard et al, 2011). And from Table 3, authors use same Vd of NH3 and NO2 for all three canopy and two countries. This methodology is not valid. Vd is affected by weather conditions and canopy types. It is changed diurnal, seasonal, and yearly for different plant types in different places. Therefore, it is not proper to European Vd data to represent Vd in China and Mongolia. Since weather data are available for authors from meteorological stations, Vd (hourly, daily or monthly) for different canopy of different Nr could be calculated easily. There are many widely used methods for this calculation, such as using big-leaf model. Response: Thanks for your suggestion. The Vd is an important parameter for the atmospheric dry and wet deposition. At the beginning of this manuscript, we read a lot of related reference to choose suitable Vd according to the similarly land use. Even so, we also according to your request, and use the model recommend by Dr. Shen (Institute of Subtropical Agriculture, the Chinese Academy of Sciences) to simulate Vd .

(3) This study used the self-made wet collection equipment as shown in Fig. 3. However, the equipment is not well designed. The top of the equipment is open without a lid, which may also collect some dry deposition especially particulate N deposition, therefore, the collected deposition by this kind of equipment is usually considered as a mixture of wet deposition and part of particulate dry deposition. By using this equipment, the wet deposition is therefore overestimated. Authors did not include any correction for this either, which resulted in high uncertainty of the reported N deposition

amount. Response: Thanks for your suggestion. Indeed, such a rain collection method will lead to a part of dry N deposition into our self-made rainfall collectors, but this was the best way can be used this kind of areas we thought, because the lack of electricity and manpower, it is difficult to collect samples after every rainfall event. Therefore, based on this, at the farmland point, we collected the samples both the self-made rainfall collectors and after every rainfall event to compare the difference and correct the result include other points. For the mountain point, very little particulate matter deposition into the grassland due to the fresh air, even so, we also collect timely rain in June to compare the difference with man-made equipment in the same month in order to make correction. So, the wet deposition data can represent the real result in our research area.

(4) The discussion needs improve. All discussion section just repeat some results and simply compared with several other studies but no detailed and deep discussion was presented. The difference of total N deposition amount and contribution of different Nr forms for different land uses could be due to the difference of fertilizer input, management, emission, and deposition process. And the difference between two countries might reflect the difference of management tradition and urban development. The discussion about Nr deposition and their potential sources among different land uses and between China and Mongolia are much more important and interesting than the comparison of the annual values with other studies since one year observation result in the manuscript is not enough to represent annual N deposition. ResponseïijŽThanks for your suggestion. It is more meaningful to compare the differences between the two places because of the difference of farming practices, population and social development levels, we will strengthen this part of the elaboration in our discussion section.

(5) The section 4.3 try to discuss the uncertainty of N deposition, but authors just discussed about NH3 compensation point. It will be better if authors could add an uncertainty analysis about N deposition, which could partly compensate the shortage of one year observation. ResponseïijŽThank you for your suggestion. we will supplement

the uncertainty of the N deposition in our discussion.

There are many format and punctuation problems in the manuscript. I suggested authors to do some technical corrections about spelling and format at the Initial Manuscript Evaluation report, but unfortunately, these errors were not corrected for the discussion paper. (1) Punctuation problems. Space is needed after each word! For instance, page 1, line 19, add a space before "at six sampling: : :"; page 1, line 24, add spaces before "in China" and before " in Mongolia". There are lots of this problem in the manuscript. Please carefully check the whole manuscript and corrected these problems! (2) Table 3, reference Shen et al. (2011) was not found in References. Please check citation and reference list. (3) Fig.1, please add Nansha Islands for China map. (4) Fig. 6, figure caption should start from the next line. (5) Fig. 7 and Fig. 8, figure captions are not consistent with figures. (6) Fig. 9, remove "liner fit of B" in figure.

ResponseïijŽThank you for your suggestion. We have carefully modified the manuscript according to your requirements.

Please also note the supplement to this comment:
http://www.biogeosciences-discuss.net/bg-2017-55/bg-2017-55-AC2-supplement.pdf

---

## Author Comment (AC1)

**Atmospheric dry and wet nitrogen deposition in agro-pastoral catchments of the China and Mongolia Altay**

Jin Ling Lv[1, 2, 3], Andreas Buerkert[4], Guo Jun Liu[1], Chao Yan Lv[1], Kai Hui Li[1] and Xue Jun Liu[1, 2*a]

[1]Xinjiang Institute of Ecology and Geography, Chinese Academy of Sciences, Urumqi 830011, China;

[2]College of Resources and Environmental Sciences, China Agricultural University, Beijing 100193, China;

[3]Institute of Plant Nutrition, Resources and Environmental Sciences, Henan Academy of Agricultural Sciences, Zhengzhou, China;

[4]Universität Kassel, Organic Plant Production and Agroecosystems Research in the Tropics and Subtropics, Steinstr. 19, D-37213 Witzenhausen, Germany

**Abstract**

Very few comparative studies on nitrogen (N) deposition in agroecosystems have been conducted along land use and altitude gradients. In an effort to fill this knowledge gap we selected three typical, interconnected land use types (cropland, mountain grassland and plain grassland) with six sampling sites in the transboundary Altay Mountains of northwest China and western Mongolia. During 12 months from June 2014 to May 2015 dry and wet N deposition, through middle volume total suspended particulates (TSP), passive samplers and precipitation collectors were monitored. Among land use types, cropland had the highest concentrations of $NH_4^+$-N (1.6 mg N $L^{-1}$ in China and 2.0 mg N $L^{-1}$ in Mongolia) and $NO_3^-$-N (1.0 mg N $L^{-1}$ in China and 1.2 mg N $L^{-1}$ in Mongolia) in precipitation compared to mountain and plain grasslands. In contrast, the Mongolian mountain grasslands (MM) experienced the high wet deposition (3.2 kg N $ha^{-1}$ $yr^{-1}$) which was at least partly due to high summer precipitation (161 mm), followed by the Chinese cropland (CC) with 3.1 kg N $ha^{-1}$

[*a]Corresponding author (X. Liu), E-mailliu310@cau.edu.cn

$yr^{-1}$ while wet deposition in other land use types ranged from 1.8 to 2.5 kg N $ha^{-1}$ $yr^{-1}$.

CC had the highest $NH_3$ (3.1µg N $m^{-3}$) and $NO_2$ (3.8µg N $m^{-3}$) concentrations and dry

N deposition (9.5 kg N $ha^{-1}yr^{-1}$) among all land use types ($p<0.05$) while Mongolian cropland (MC) had dry N deposition of 5.4 kg N $ha^{-1}yr^{-1}$. CC (12.6 kg N $ha^{-1}$ $yr^{-1}$) had the highest total N deposition, followed by the MC with 7.2 kg N $ha^{-1}$ $yr^{-1}$ and the

Mongolian mountain grassland (MM) with 6.6 kg N $ha^{-1}$ $yr^{-1}$. $NH_4^+$-N concentration in the precipitation were negatively correlated with precipitation ($P<0.05$).

Concentration of $NH_3$ correlated positively with air temperature ($P<0.05$) probably reflecting promoting effects of temperature on $NH_3$ emissions whereas $NO_2$ correlated negatively with temperature due to low background value of $NO_2$ emission. Overall, croplands in China had 76 % higher N deposition than in Mongolia whereas the reverse was true for mountain grasslands which received 26 % more N in Mongolia.

**Key words:** Agro-pastoral transition zone; Dry deposition; Land-use types;

Transborder watershed; Wet deposition

**Introduction**

During the last three decades China´s industrial development and intensification of agriculture and livestock production have greatly increased the concentration and deposition of atmospheric reactive nitrogen (Liu et al., 2013). Since the 1980s, atmospheric Nr emissions have more than doubled in north and southeast China following 30 years of strong economic development (Liu et al. 2013) and current atmospheric Nr deposition are still very high (Pan et al., 2012; Xu et al. 2015). Total nitrogen deposition of 54.4-103.2 kg N ha$^{-1}$ yr$^{-1}$ have been reported (Luo et al. 2013) in the North China Plain and up to 35 kg N ha$^{-1}$yr$^{-1}$ in a remote oasis area of Xinjiang, northwest China (Liu et al. 2011; Li et al. 2012) .

Rapid intensification of agriculture in response to the increasing demand for food has led to increased use of mineral fertilizers in China which is one of the main factors responsible for high regional Nr deposition (Ju et al. 2009; Zhang et al. 2013). Also along with the increase of transport, animal breeding and energy extraction, more pollutants swarm into the atmosphere and constitute the main source of atmospheric wet and dry nitrogen deposition. Therefore, there is a complex set of circumstances for the distribution of atmospheric nitrogen deposition in China due to the huge regional differences and rapid economic development (Huang et al. 2013; Wang et al. 2010).

Previous N deposition studies have mainly focused on agricultural and urban areas (Luo et al. 2003; Li et al. 2012; Du et al. 2014; Huang et al. 2015; Liu et al. 2015; Wasiuta et al. 2015). Little research has been conducted to quantify atmospheric N deposition in agro-pastoral zones. Currently approximately 117 million people depend on this land use system which are mainly distributed in the northeastern and northwestern part of the country and in Inner Mongolia (Du et al. 2009; Xu et al. 2014; Zhang et al. 2015) . Grasslands within this agro-pastoral zone in China have been experiencing the use of high amounts of chemical fertilization and overgrazing due to high livestock numbers where atmospheric N deposition may be interacting under such conditions. So, the monitoring of atmospheric dry and wet N deposition is therefore urgently needed in this region. This can provide a basis for a more sustainable utilization of nitrogen fertilizer in farmland and the improvement of deteriorated grassland.

Additionally, the different levels of socio-economic development, cropland management practices and animal husbandry systems could also influence the dry and wet N deposition. However, very few comparative studies for dry and wet N

deposition have been conducted in these two regions which have different levels of development. So, the present study was carried out in the Altay Mountains, near the border area of northwest China and western Mongolia which represent a typical agro-pastoral transition zone. The two study areas have similar land use types but different cropland/grazing land ratios, intensities of mineral fertilizer input, animal husbandry systems and levels of urban development due to the different historical and cultural background as well as the significant political and economic transformation processes(Greta et al., 2016). As the combination of these factors may lead to significant differences in N deposition over time, which could also result in a meaningful guidance for the sustainable utilization of nitrogen in both countries, our aims were to quantify seasonal variations in atmospheric N deposition and to compare the difference of dry and wet N deposition in China and Mongolia agro-pastoral catchments.

**2. Materials and methods**

**2.1. Sampling sites**

The study was conducted in Qinghe and Bulgan counties located in northwest China and Mongolia respectively, comprising an area ranging from 45-47 °N and 89-91 °E. The topography is characterized by a gradual decline in elevation from north to south and is divided into high, intermediate and low mountains, hills, and the Gobi desert zone. The average altitude of Qinghe county is 1218m above sea level (a.s.l.), with a maximum elevation of 3659 m a.s.l. and a minimum of 900m a.s.l.. Qinghe county is situated in the continental north temperate arid climatic zone. Air humidity is very low all year round with an average annual precipitation of 161 mm and an annual potential evaporation of 1495 mm. Winters are long and cold with an absolute minimum of -53 °C followed by cool and short summers with a recorded maximum temperature of 36.5 °C and an average air temperature of 0 °C (Fig. 2). The pasture area in Qinghe comprises about 14300 km$^2$ with a cultivated land area of 0.126 million hectares and 1.64 million farm animals traded annually. Most of the agricultural area is cultivated with spring wheat (*Triticumaestivum* L.), though alfalfa (*Medicago sativa* L.) and sea buckthorn (*Hippophaerhamnoides* L.) are also widely grown. Wheat is sown in early May and harvested at the beginning of September. The amount of mineral fertilizer applied to wheat ranges from 300-350 kg N ha$^{-1}$ yr$^{-1}$ in the Chinese croplands. Large numbers of sheep, cattle and camels are moved into the mountain grassland (summer pasture) from July to September while during the winter months they remain for stubble grazing in the oasis croplands and plain grassland (winter pasture).

Bulgan county (or Soum) neighbors the Bayan-Olgii Province in the north, Northwest China in the east, and Uyench and Altay soum of Khovd province in the south. Its territory comprises 8105 km$^2$ at an average altitude of 1164m a.s.l.. It has a continental climate with four seasons: April and May are the windiest months, January is the coldest (-40 °C) and July is the warmest (+35 °C) month of the year.

Annual precipitation averages 120-140 mm, with pronounced spring and autumn seasons allowing some cultivation of arable crops. Most of the cropland of Bulgan soum is also planted with spring wheat and to a lesser extent with rye (*Secalecereale*L.) with 150-250 N kg ha$^{-1}$ yr$^{-1}$. Similarly to Qinghe, the growing season extends from May to September and large numbers of sheep, cattle and camels are moved into the mountain grassland from July to September to spend the winters in the lowlands.

In addition, the three similar land use types that were investigated (six sampling sites) are described as follows: Chinese cropland (CC), Chinese mountain grassland (CM), Chinese plain grassland (CP), Mongolian cropland (MC), Mongolian mountain grassland (MM) and Mongolian plain grassland (MP).

**2.2. Measurement of N deposition and analytical procedures**

From June 2014 to May 2015, wet (i.e. bulk) and dry N concentrations and deposition were monitored and quantified at six sites in the border area of Altay Mountains(Fig.

1).

*2.2.1 Rainwater collection and calculation of wet N deposition*

Rainwater samples were collected with precipitation collectors directly after every rainfall event in Chinese (CC) and Mongolian (MC) croplands by local farmers and/or herdsmen and the dates and amounts of rainwater recorded. Also, some special rainfall collectors (three sub-samples per site) were employed at, mountain grasslands and plain grasslands in both China and Mongolia due to the difficulty of collecting samples after every rainfall event. Same equipments were deployed in the cropland site in order to compare values against rainfall events in a month. For the cropland and mountain grassland, we collected rainfall or snow samples every month throughout the year. For the plain grassland, however, we just collected samples every month during the growing season(from May to October). Outside the growing season samples were collected just for the first time and last snowfall event to compute average values. Each rain collector comprised a funnel container of 40 cm in diameter, a plastic hose and a 20 Liter capacity plastic bucket. The funnel container was set at approx 1.5 m above the ground to avoid dust or leaf contamination from the ground surface. The plastic kettles were grounded at 30 cm depth, stick out 5cm of the ground and were covered with a lid to prevent entry of dust or other pollutants connected to the funnel with a plastic hose. Chloroform ($CHCl_3$, 1-2 ml) was added to each bucket to inhibit N transformations in the rainwater samples. The amount of precipitation was measured by an automatic weather station located close to the Meteorological

Bureau of Qinghe Country. The precipitation samples were collected manually once per month at the six sampling points and transferred to plastic bottles and subsequently stored in a refrigerator at -10 °C until analysis. We also sampled snow at the beginning and end of the snowfall period to combine the snowfall data and calculate winter wet N deposition. All samples were analyzed for $NH_4^+$-N and $NO_3^-$-N

(inorganic N) concentrations using an AA3 continuous flow analyzer (Seal Analytical

Ltd., Southampton, UK). Wet deposition of inorganic N was calculated according to

Luo et al. (2014) as follows:

Wet N (every rainfall event, kg N ha$^{-1}$) = inorganic N concentration (mg N L$^{-1}$)

$\times$precipitation (mm) $\times$0.01

Wet N (every month, kg N ha$^{-1}$) = 0.001$\times\Sigma$  N (every rainfall event or month)

*2.2.2 NH$_3$ and NO$_2$ collection and N calculation*

Atmospheric $NO_2$ was collected with passive samplers using Gradko diffusion tubes (Goulding et al., 1998). The $NO_2$ samplers consisted of polyethylene tubes (71.0 mm long and 11.0 mm internal diameter) with two caps and stainless steel mesh disks.

Two dry disks were placed in the caps and 30 ml of a 20% aqueous solution of triethanolamine was pipetted into the gray cap. The samplers were suspended at 1.5 m (at least 0.5 m higher than the canopy height) above ground and exposed between 15

and 30 days in the air every month. The disks were extracted with a solution containing      sulphanilamide,      $H_3PO_4$      and      N-1-naphthylethylene-diamine dihydrochloride to estimate the $NO_2$ concentration determined by colorimetry at a wavelength of 542 nm. $NH_3$ samples were collected using ALPHA passive samplers (Adapted Low-cost High Absorption, Center for Ecology and Hydrology, Edinburgh,

UK). This equipment included a tube, a plastic filter and a membrane (absorbed citric acid) and was placed   approximately 1.5 m above ground. For the $NO_2$ and $NH_3$, we collected the samples one month one time. The cropland and mountain grassland samples were collected from June 1, 2014 to May 31, 2015, and the plain grassland samples were just collected in growing season due to harsh environmental conditions.

The calculation was made according to Luo et al. (2014) as follows:

 $V=DAt/L$

Where t represents the time interval; $D=2.09 \times 10^{-5} \, m^{-2} \, s^{-1}$ at 10°C, $A=3.463 \times 10^{-4} \, m^{-2}$,

$L=0.006$ m. The following equation was then derived:

 $V(m^3) = 0.004343363(m^3) \times t(h)$

 The concentration of $NH_3$ ($\mu g \, N \, m^{-3}$) was obtained as follow:

 $C= (m_e - m_b)/V$

Where $m_e$ represents the amount of $NH_3$ in the experimental sample and $m_b$ represents the amount of $NH_3$ in the blank sample.

*2.2.3  $pNH_4^+$ and $pNO_3^-$*

Airborne $PM_{10}$ particles (particulate matter with a aerodynamic equivalent diameter of

< 10 μm) were sampled using a middle flow particulate sampler (Tian hong

Instruments Co. Ltd., Wuhan, China) with a flow fluxes of $1.05 \, m^3 \, min^{-1}$. Seven to ten daily samples of $PM_{10}$ were collected at QC and BC during each month. Samples from the other sites were not taken due to lack of electrical power and harsh environmental conditions.

The membrane of $PM_{10}$ consisted of a glass fiber and it was placed in an incubator at constant temperature and humidity (22°C, relative humidity 50%) for 24 hours before and after sampling and weighed on an electronic balance. Finally, the samples were placed in beakers containing 50ml ultrapure water and ultrasonicated for 30 min.

The extracts were filtered through 47-mm Whatman GF/F membrane syringe filters (GE Healthcare Bio-Sciences, Pittsburg, PA, USA). The filtrates were stored and refrigerated at 4 °C. Ammonium and nitrate in $PM_{10}$ ($pNH_4^+$ and $pNO_3^-$) were measured using a Seal AA3 continuous flow analyzer (Seal Analytical Ltd.,

Southampton, UK).

**2.3 Estimation of dry N deposition**

The effects of changing weather conditions and differences in vegetation types make the collection of Data on dry N deposition complicated as(Yu et al. 2014; Simpson et al., 2014). So, we used micro-meteorological methods and estimated dry N deposition by multiplying the measured concentrations of Nr species by their deposition velocities ($V_d$) in our experimental sites. The $V_d$ of $NH_3$, $NO_2$ and TSP can be calculated in accordance with the method recommend by Shen (Shen et al. 2013). The following equations were used:

$F=C \times V_d$

Where by $V_d$ can be expressed by

$V_d = (R_a + R_b + R_c)^{-1}$

Where $R_a$ is the aerodynamic resistance, $R_b$ is the quasi-laminar boundary layer resistance, and $R_c$ is the surface or canopy resistance.

**2.4 Statistical analysis**

Linear regression was used to analyze interactions among the different Nr species. For

Pearson ś correlation and linear regression analyses, significance was defined at

$P<0.05$. T-tests and one way analysis of variance (ANOVA) were employed to compare N deposition among monitoring sites, land-use types and seasons. All statistical analyses were performed using the SPSS 18.0 software package (SPSS Inc.,

Chicago, IL, USA). Figures were prepared using the Origin 8.0 software package (Origin Lab Corporation, Northampton, MA, USA).

**3 Results**

**3.1 Wet deposition of $NH_4^+$-N and $NO_3^-$N**

The $NH_4$+-N concentration in rainwater collected from MC was highest compared to the MM and MP (Table 2, Fig. 3).The $NH_4^+$-N concentrations of the samples from

Chinese sites were relatively low compared to the Mongolian sampling sites. CC had relatively higher $NH_4^+$-N concentration compared to CM and CP. $NO_3^-$-N

concentrations were highest for CP in China, followed by MC in Mongolia.

The different land use types had different $NH_4^+$-N and $NO_3^-$-N peaks. Highest

$NH_4^+$-N concentration occurred in May in CC and in September in MC and the highest $NO_3^-$-N peak occurred from March to May in CC and from August to October in MC (Fig.3). Except the cropland, the $NO_3^-$-N peaks were recorded from July to

[revised manuscript text omitted]

deposition (5.7 kg N ha$^{-1}$) compared to CP grassland (5.7 kg N ha$^{-1}$). In addition, the wet N deposition species ($NH_4^+$ and $NO_3^-$) altogether accounted for 25-48% at the

Chinese sites and 25-49 % at the Mongolian sites. Dry N deposition accounted for

52-75% at the Chinese sites and 51-75% at Mongolian sites.

**4 Discussion**

**4.1 Atmospheric dry and wet N deposition**

Dry deposition includes gas emissions and particulate Nr deposition (Shen et al. 2013;

Granath et al. 2014; Maaroufi et al. 2015). In our experiment CC had significant higher $NH_3$ and $NO_2$ concentration than the other land use types, mainly due to the large area of cropland on the Chinese side of the border, together with the excessive inputs of mineral N fertilizer, which likely led to large losses *via* $NH_3$ volatilization and soil NOx emissions. Compared to MM, we found that the CM had a significant higher $NH_3$ concentration from June to September than MM, which was mainly due to more livestock and excrements in the mountain per unit area (p<0.05). However,

$NH_3$ concentration showed the opposite trend in winter in MC, which could have been a as result of having more livestock numbers staying in the Mongolian mountain in winter time due to the difference of traditional grazing practices. Except for $NH_3$

deposition, the MM had higher $NO_2$ depositions than the CM grassland, probably as a consequence of many herdsmen staying in the Mongolian mountain site, especially in winter when large amounts of coal, wood and cattle manure are burned for home heating from October to May. However, herdsmen in China move to the mountains only from July to mid-September in summer, with very few people living there during the winter. A similar explanation could be suggested for the wet deposition.

The monthly concentrations of $NH_3$ showed significant positive correlations with temperature (P=0.009) but no correlation with either RH (P=0.491) or $NO_2$ (P=0.580;

Fig.10).This result was consistent with other findings, for example, a similar trend was also found in Guangzhou in south China and in an agricultural catchment in subtropical central China (Yang et al. 2010; Shen et al. 2013). This suggests that increasing temperature promotes the emission of $NH_3$. Gaseous $NO_2$ was also positive correlated with temperature but neither with RH nor $NH_3$. This may also imply that

$NO_2$ emissions mostly occur as a consequence of human activities, especially the combustion of fossil fuels and automobile exhausts with similarly results in other places but a relatively low value in agro-pastoral areas. The amount of rainfall had a significant effect on the concentration of inorganic N. The higher amount of precipitation, the lower the inorganic N concentration, especially in the case of $NH_4^+$

which was significantly correlated with the precipitation (P=0.039). $NH_4^+$ and

$NO_3^-$ were not significantly correlated with one another (P=0.143), which indicated that the results for the wet deposition are greatly influenced by the dry deposition in our research area. All in all, the different land use types did not differ significantly in their wet deposition in either country.

**4.2 Effects of N deposition on N cycling in the agro-pastoral transition zone**

**between China and Mongolia**

The effects of N deposition in the agro-pastoral transition zone are different from other areas. Nitrogen deposition in agriculture areas is mainly affected by fertilizer application rates (Li et al. 2012). However, in the agro-pastoral transition zone, a part of fertilizer applications, the seasonal migration of livestock (transhumance) leads to the general distribution of large amounts of animal manures and which represents the second N emission source. So in this study we found that the dry deposition had higher percentage than wet deposition in agro-pastoral catchments in cropland.

However, a different result occurred in the mountain grassland with almost equal proportions for wet and dry deposition which may be explained by higher rainfall with low $NH_3$ and $NO_2$ concentrations in mountain grassland and relatively lower rainfall with higher $NH_3$ and $NO_2$ concentrations in cropland in this catchment. The different proportion of atmospheric wet and dry N deposition in mountain grassland and cropland appears to be the important feature in agro-pastoral area with different altitudinal gradients.

Furthermore, as well as other studies (Aber et al. 1997; Flechard et al., 2011;

Azati et al. 2014), we found that the grasslands face serious N losses especially in mountain areas where large numbers of grazing animals move into the mountain grassland and remain for 3-4 months leading to substantial loss of nutrients under long-term migration conditions. In China, during the last decades, in parts of the

Altay- Dzungarian region, laws and policies were implemented to intensify livestock production while reducing the rate of land degradation (Greta et al., 2016), however, the overgrazing phenomenon is also very common, which still leads to the loss of nutrients with livestock being transported to other cities. Instead in Mongolia a larger proportion of the meat is consumed locally and the majority of livestock continue to graze in the mountains throughout the year, leading to more closed N cycles in grassland areas. Thus, N cycles in grasslands are different in China and Mongolia, with Chinese grasslands facing a more pronounced risk of N losses. Our study shows that N deposition in cropland differs between Mongolia and China, mainly due to higher application rates of fertilizer N in China. Therefore China and Mongolia exhibit different N deposition and N cycling rates reflecting different land use management intensities, grazing systems and trading conditions.

**4.3 The uncertainty of the compensation point between the $NH_3$ emission and**

**deposition**

The concentration of $NH_3$ in the air is susceptible to be affected by meteorological and anthropogenic factors. On the one hand, part of atmospheric ammonia settled onto the soil surface, And part of $NH_3$ volatilize from the surface soil. Therefore, it is difficult to accurately estimate net $NH_3$ deposition under the conditions of this study.

In order to better estimate the $NH_3$ deposition value, it is common practice to calculate the deposition velocity rate by means of meteorological factors to get the appropriate deposition compensation point. In our study, the land use included mountain grassland (alpine meadow), plain grassland and farmland. In the farmland, 5.0μg N

$m^{-3}$ was assumed as the compensation point of dry deposition of $NH_3$ in the growing season (Shen et al. 2013), and 0 μg N $m^{-3}$ was assumed as the compensation point of dry deposition of $NH_3$ in the no-growing season due to low $NH_3$ volatilization. In the mountain and plain grassland, 0 μg N $m^{-3}$ was chosen as the compensation point of dry deposition of $NH_3$ due to low $NH_3$ volatilization (Li et al., 2012; Shen et al. 2013).

Except the $NH_3$ compensation point, the value of wet and dry deposition for different land use styles had the interannual variation due to the change of cropland area and number of livestock with climatic variation in local area. Beyond that, we observed that N deposition was spatially very unevenly distributed, particularly between mountain pastures and plain pastures. Nitrogen deposition was possibly higher next to herdsmen's houses, roads or sheepfolds due to more pronounced $NH_3$ or $NO_X$ releases.

Farm- and grasslands are intertwined in our research areas. Therefore, much uncertainly for wet and dry N deposition remain.

**5 Conclusions**

The agro-pastoral area around Qinghe (China) and Bulgan (Mongolia) differed in atmospheric N deposition across land use types. The mountain grasslands had relatively higher wet deposition reflecting much higher rainfall and Nr emissions.

Chinese croplands had higher wet and total N deposition than Mongolian croplands due to higher population and chemical fertilizer input, but higher N deposition were found in the Mongolian mountain grassland than Chinese mountain grassland due to different grazing systems. Nearly all land use types had higher N deposition in the (warm) growing season than in the winter months. Compared to Mongolia, Chinese grassland faces more pronounced Nr losses due to additional N deposition and overgrazing, suggesting that a reduction of the application of N-fertilizers to croplands as well as livestock numbers would help to decrease N deposition.

**Acknowledgements**

We acknowledge Dr. Peter Christie (UK) and Dr. Olave Rodrigo for his valuable comments and linguistic corrections of the manuscript. We also thank Dr. Sven Goenster (Universität Kassel, Germany) for his contribution providing meteorological data and part of sample collection and Dr. Jianlin Shen for his contribution using the meteorological data to simulate the $V_d$ value of different land use types. The study was supported by the WATERCOPE (I-R-1284-WATERCOPE) project funded by IFAD (International Funding for Agriculture Development, Rome, Italy), the State Basic Research Program (2014CB954200) and the Chinese National Natural Science Foundation (41425007, 31421092).

[revised manuscript text omitted]

[a] Dry deposition velocities of NH$_3$ was 0.4, 0.55 and 0.42 cm s$^{-1}$ for CM, CC and CP, 0.41, 0.52 and 0.41 cm s$^{-1}$ for MM, MC and

MP respectively. Dry deposition velocities of NO$_2$ was 0,26, 0.3 and 0.26 cm s$^{-1}$ for CM, CC and CP, respectively. 0.24, 0.26 and

0.26 cm s$^{-1}$ for MM,MC and MP respectively. Dry deposition velocities of pNH$_4^+$ and pNO$_3^-$ was 0.2-0.22 cm s$^{-1}$ for CC and

0.20- 0.22 cm s$^{-1}$ for MC, the method was from Shen et al.(2013)

[b] WD: total wet N deposition, DD: total dry N deposition, TD: total N deposition

[Figure]

**Fig.1.** Map of the six sampling sites in the agro-pastoral catchment of the Chinese and Mongolian Altay Mountains.

[Figure]

**Fig. 2.**Monthly mean air temperature and relatively humidity (RH) at six sampling sites of the Chinese and Mongolian Altay Mountains.

**Fig. 3.**The self-made wet collection equipment at the sampling sites in the Chinese (up right) and Mongolian Altay Mountains (down right)

[Figure]

**Fig.4.**Concentration of $NH_4^+$-N and $NO_3^-$-N of wet deposition at six samples sites in the Chinese and Mongolian Altay Mountains

[Figure]

**Fig.5.**Monthly concentrations of $NH_3$-Nin the growing season (G) and the non-growing season (NG) at six sites in the Chinese and Mongolian Altay Mountains

[Figure]

**Fig.6.** Monthly concentrations of NO$_2$-N in the growing season (G) and the non-growing season (NG)of six sites in the Chinese and Mongolian Altay Mountains

[Figure]

**Fig.7.**Monthly concentrations of $NO_2$-N at six sampling sites in Chinese and

Mongolian Altay Mountains

**Fig.8.**Concentrations of $NO_2$-N in the G (growing season) and the NG (non-growing season) at six sites in the Chinese and Mongolian Altay Mountains

[Figure]

**Fig.9.** Relationship between monthly precipitation and $NH_4^+$-N and $NO_3^-$-N in rainwater at six sampling sites in the Chinese and Mongolian Altay Mountains

[Figure]

**Fig.10.** Relationship between atmospheric NH₃ and NO₂ and temperature (Temp) and relatively humidity (RH) in the Chinese and Mongolian Altay Mountains.